# Neuronal Depolarization Induced RNA m^5^C Methylation Changes in Mouse Cortical Neurons

**DOI:** 10.3390/biology11070988

**Published:** 2022-06-29

**Authors:** Xiguang Xu, Zachary Johnson, Hehuang Xie

**Affiliations:** 1Epigenomics and Computational Biology Lab, Fralin Life Sciences Institute, Virginia Tech, Blacksburg, VA 24061, USA; xiguang@vt.edu (X.X.); zaustinj@vt.edu (Z.J.); 2Department of Biological Sciences, College of Science, Virginia Tech, Blacksburg, VA 24061, USA; 3Genetics, Bioinformatics and Computational Biology Program, Virginia Tech, Blacksburg, VA 24061, USA; 4Department of Biomedical Sciences and Pathobiology, Virginia-Maryland College of Veterinary Medicine, Virginia Tech, Blacksburg, VA 24061, USA; 5Translational Biology, Medicine and Health Program, Virginia Tech, Blacksburg, VA 24061, USA

**Keywords:** neuronal depolarization, gene expression, RNA cytosine-5 methylation, RNA bisulfite sequencing, RNA-seq

## Abstract

**Simple Summary:**

When neurons are activated, two waves of gene expression changes occur to achieve neuronal adaptation to their environment. Currently, little is known about the changes in post-transcriptional modifications of neuronal transcripts in response to environmental stimuli. This study aims to investigate the dynamics of m^5^C modification on neuronal mRNAs upon depolarization using potassium chloride. We found that neuronal mRNAs carrying m^5^C modifications are enriched for genes with important neuronal functions. The numbers of m^5^C sites and methylated transcripts increased at 2 h after neuronal depolarization and slightly dropped at 6 h. Interestingly, for differentially methylated genes, a negative correlation between the levels of gene expression and RNA methylation is more prevalent. We anticipate our findings may shed new lights on post-transcriptional regulation in activated neurons, and the transcriptomic plus epi-transcriptomic datasets generated in this study would provide a valuable resource to the scientific community.

**Abstract:**

Neuronal activity is accomplished via substantial changes in gene expression, which may be accompanied by post-transcriptional modifications including RNA cytosine-5 methylation (m^5^C). Despite several reports on the transcriptome profiling of activated neurons, the dynamics of neuronal mRNA m^5^C modification in response to environmental stimuli has not been explored. Here, we provide transcriptome-wide maps of m^5^C modification, together with gene expression profiles, for mouse cortical neurons at 0 h, 2 h, and 6 h upon membrane depolarization. Thousands of differentially expressed genes (DEGs) were identified during the neuronal depolarization process. In stimulated neurons, the majority of early response genes were found to serve as expression regulators of late response genes, which are involved in signaling pathways and diverse synaptic functions. With RNA bisulfite sequencing data, a union set of 439 m^5^C sites was identified with high confidence, and approximately 30% of them were shared by neurons at all three time points. Interestingly, over 41% of the m^5^C sites showed increased methylation upon neuronal activation and were enriched in transcripts coding for proteins with synaptic functions. In addition, a modest negative correlation was observed between RNA expression and methylation. In summary, our study provided dynamic transcriptome-wide landscapes of RNA m^5^C methylation in neurons, and revealed that mRNA m^5^C methylation is associated with the regulation of gene expression.

## 1. Background

Post-transcriptional modification of RNA is emerging as a new layer in the regulation of gene expression [1]. With recent advances in biochemical detection techniques, researchers have identified more than 170 types of RNA modifications [2], including cytosine-5 methylation (m^5^C). RNA m^5^C modification was first identified in the more abundant and relatively stable ribosomal RNA and transfer RNA [3,4]. With next generation sequencing (NGS) techniques, m^5^C modification was found to be widely present in messenger RNA (mRNA) and non-coding RNA [5,6]. The m^5^C modification in mRNA is introduced mainly by NOP2/Sun RNA (NSUN) methyltransferase family members [7,8]. Removal of the m^5^C modification in RNA was mediated by three ten-eleven translocation enzymes (TET1/TET2/TET3) through a sequential oxidization process to form 5-hydroxymethylcytosine (hm^5^C), 5-formylcytosine and 5-carboxylcytosine [9,10,11]. However, *Tet1/Tet2/Tet3* triple knockout mouse embryonic stem cells (ESCs) showed reduced but detectable RNA hm^5^C levels compared to wild-type ESCs [9], indicating that additional unknown enzymes may be involved in the RNA demethylation pathway.

Despite the elusive pathway for RNA m^5^C demethylation, recent studies revealed critical roles for m^5^C modification in RNA metabolism. With aid from the m^5^C reader protein ALY/REF export factor (ALYREF), m^5^C in mRNAs facilitates mRNA export from the nucleus to the cytoplasm [7]. Transcriptome-wide mapping of m^5^C modification shows a significant enrichment in the vicinity of the translational start sites and the 3′-untranslated regions (3′UTRs), suggesting that m^5^C plays a role in regulating mRNA translation and stability [5,7,12]. mRNA stabilization is achieved via binding of an m^5^C reader protein Y box binding protein 1 (YBX1) in zebrafish and Ypsilon schachtel (YPS) in Drosophila [13,14]. Interestingly, regulation of the m^5^C modification in mRNAs is highly dynamic and was found to be involved in diverse physiological and pathological conditions, including the regulation of mouse testis tissue development [7], development of ovarian germline stem cells in Drosophila [13], and maternal-to-zygotic transition (MZT) in Zebrafish [14]. In addition, aberrant m^5^C in mRNAs has been associated with the pathogenesis of human bladder cancer [15].

In the nervous system, a number of studies focused on the functional understanding of NSUN2, an m^5^C RNA methyltransferase. Genetic mutations in the NSUN2 gene are associated with Dubowitz syndrome and intellectual disability [16,17]. NSUN2 catalyzes cytosine methylation of tRNA precursors to stabilize the anticodon-codon pairing. The loss of NSUN2 leads to a decrease in tRNA m^5^C methylation and the accumulation of tRNA fragments, which reduces translation efficiency and increases neuronal apoptosis in response to stress [18]. Aberrant tRNA m^5^C methylation contributes to the deficiency in migration and differentiation of neural stem cells, impaired synaptic plasticity in brain, and disordered complex behaviors [19,20]. A few recent studies suggest that m^5^C methylation in mRNAs may also play a role in brain tissue development and function. Compared with those of mouse embryonic stem cells, methylated transcripts in the brain have distinct m^5^C profiles and are enriched in genes with ion transport or synapse function [12]. In glioblastoma cells treated with alkylating agents, the m^5^C methylation of transcripts mediated by NSUN6 may regulate transcriptional and translational processes [21]. Using a model for neuronal oxygen-glucose deprivation/reoxygenation, Jian et al. linked neuronal mRNA m^5^C methylation to the stress response and apoptosis regulation [22]. Currently, no study has been performed to examine the link between mRNA m^5^C methylation and neuronal activity.

Neuronal activity-driven gene expression is essential for neurons in response to environmental stimuli. Such changes in the neuronal transcriptome may lead to long-lasting structural and electrophysiological adaptations in the neural circuit during development, learning and memory formation [23,24]. In response to stimulation, previous studies reported dynamic changes in DNA methylation and chromatin accessibility in neurons [25,26]. Changes in RNA cytosine-5 methylation in activated neurons has not been studied yet. To investigate the dynamic changes of m^5^C modification, we adopted a widely used model mimicking neuronal activity, in which the in vitro cultured mouse cortical neurons were depolarized with potassium chloride [27]. Membrane depolarization triggers a calcium influx and activates a complex signaling cascade with highly dynamic gene expression [27,28]. In this study, we took advantage of such an ideal system to investigate the dynamics of m^5^C modification in neurons in response to environmental stimuli and explore the correlation between the changes in gene expression and RNA methylation.

## 2. Materials and Methods

### 2.1. Animal

C57BL/6 mice were maintained and bred in a 12 h light/dark cycle under standard pathogen-free conditions. Adult female and male mice were used for time pregnancy by checking vaginal plugs daily in the morning. Positive plugs were designated as E0.5.

### 2.2. Primary Mouse Cortical Neuronal Culture

Primary mouse cortical neurons were prepared as previously described [27] with some modifications. Briefly, C57BL/6 E16.5 mouse embryos were micro-dissected for cortex tissues and the cortex tissues were dissociated into a single-cell suspension using the Neural tissue dissociation kit (P) (Cat# 130-092-628, Miltenyi Biotec, Gaithersburg, MD, USA) according to the manufacturer’s instructions. After dissociation, neuronal cells were filtered through a 70 μm strainer (Falcon, Titusville, FL, USA), and spun at 300 g for 10 min. The cell pellet was resuspended in neuronal culture medium (Neurobasal medium containing 2% B27 supplement (Invitrogen, Wilmington, NC, USA), 1% Glutamax (Thermo Fisher, Wilmington, NC, USA) and 1% penicillin-streptomycin (Thermo Fisher, Wilmington, NC, USA) and seeded on laminin and poly-ornithine coated 10 cm dishes. Neurons were grown in vitro for 7 days with fresh medium changed on day 3 and 6.

### 2.3. Membrane Depolarization and RNA Isolation

After being cultured in vitro for 6 days, neuronal cells were silenced with 1 μM tetrodotoxin (TTX; Fisher, Clarion County, PA, USA) and 100 μM DL-2-amino-5-phosphopentanoic acid (DL-AP5; Fisher) overnight. The next morning, neuronal cells were depolarized with 55 mM KCl for 0 h, 2 h, and 6 h. At the end time point, the neuronal cells were harvested and lysed with TRIzol reagent. Total RNA was extracted using TRIzol reagent combined with the RNeasy min kit (QIAGEN, Germantown, MD, USA) with DNase I on-column digestion. To enrich poly(A)-containing mRNAs, two rounds of poly(A) selection were performed using oligo(dT) beads (Thermo Fisher) following the manufacturer’s instructions.

### 2.4. Immunostaining

Immunostaining was performed as previously described [29]. Briefly, E16.5 mouse cortical neurons were dissociated and seeded on an 8-well chamber and cultured in vitro for 7 days (DIV7). The neurons were fixed with 4% paraformaldehyde in phosphate buffered saline (PBS) for 15 min and permeabilized with 0.2% TritonX-100 in PBS for 10 min. After being blocked with 5% Normal Goat Serum (Thermo Fisher) at room temperature (RT) for 1 h, the cells were incubated with mouse anti-Tuj1 antibody (Cat# 801201, Biolegend, San Diego, CA, USA) and rabbit anti-GFAP antibody (Sigma, New York, NY, USA, HPA056030) at 4 °C overnight. Then the cells were incubated with Cy3 conjugated anti-rabbit IgG (A10520, Invitrogen) and Alexa Fluor 488 conjugated anti-mouse IgG (A10680, Invitrogen) secondary antibodies at RT in darkness for 1 h. After washing 3 × 5 min with 1 × PBS, cells were then mounted with DAPI-Fluoromount-G™ Clear Mounting Media (Cat# 010020, Southern Biotech, Birmingham, AL, USA). Fluorescent images were acquired using a confocal microscope.

### 2.5. RNA-seq Library Construction

Stranded RNA-seq libraries were constructed using the TruSeq Stranded mRNA Library Preparation Kit (Illumina, San Diego, CA, USA) following manufacturer’s instructions. Briefly, after two rounds of poly(A) selection, the mRNA samples were fragmented and primed to synthesize first strand cDNA, followed by synthesis of the second strand cDNA. After Ampure XP beads purification, dA tailing was performed and indexed adapters were ligated to both ends of the double stranded cDNA. Adapter-ligated DNA fragments were enriched by PCR amplification for 12 cycles. After Ampure XP beads purification, the PCR products were size-selected in a range from 350 bp to 550 bp on 2% dye-free agarose gel using the pippin recovery system (Sage Science, Beverly, MA, USA). The recovered libraries were sequenced on a Hiseq 4000 platform in the 150 bp paired end mode (Illumina, San Diego, CA, USA).

### 2.6. RNA-seq Data Analysis

Raw reads were trimmed to remove the adapters and low-quality bases (Q < 30) using Trim Galore (https://www.bioinformatics.babraham.ac.uk/projects/trim_galore/) (accessed on 5 May 2021). Processed reads with lengths greater than 30 nt were defined as clean reads. Clean reads were mapped to the mm10 genome and gene expression level are calculated by RSEM [30]. The raw counts were used to identify differentially expressed genes by DESeq2 [31]. The criteria of differentially expression genes includes: (1) the adjusted *p*-value is less than or equal to 0.05, and (2) the gene expression fold change is greater than 2. Gene ontology (GO) analysis was performed using the DAVID online tool (2021 Update) [32]. Default parameters were used for the enrichment analysis for Biological Process (BP), cellular component (CC), and molecular function (MF).

### 2.7. Generation of Spike-In Unmethylated mRNA Control

The spiked-in unmethylated mRNA was transcribed from the pTRI-Xef plasmid supplied by the MEGAscript™ T7 Transcription Kit (Invitorgen). Briefly, the linearized pTRI-Xef plasmid was transcribed in vitro in a reaction with MEGAscript T7 RNA polymerase (Ambion, Carlsbad, CA, USA) at 37 °C for 4 h, followed by DNase treatment to remove the DNA template. The RNA sample was purified using the RNeasy Mini Kit (QIAGEN, Germantown, MD, USA). The in vitro transcribed unmethylated mRNA control was spiked at a ratio of 0.5% in the RNA samples before bisulfite treatment.

### 2.8. RNA BS-seq Library Construction

RNA bisulfite conversion was performed as previously described [12] with minor modifications. Briefly, poly(A) RNA was spiked-in with Xef unmethylated RNA and bisulfte converted using the EZ RNA methylation Kit (Zymo Research, Irvine, CA, USA) with initial denaturation at 95 °C for 1 min, followed by three cycles of 70 °C for 10 min and 64 °C for 45 min. Binding, desulphonation, and purification were performed on-column following the manufacturer’s instructions. The eluted RNA was used for stranded RNA-seq library construction using the TruSeq Stranded mRNA Library Preparation Kit (Illumina, San Diego, CA, USA) with a supplement of ACT random hexamers during first strand cDNA synthesis.

### 2.9. RNA BS-seq Data Processing

RNA BS-seq data analysis was performed as previously described [33]. Mouse transcriptome (GRCm38) and annotation files were downloaded from the Ensemble database. Raw reads were trimmed to remove the first 6 bases on the 5′ end, adapters, and low-quality bases using Trim Galore. The processed reads with lengths greater than 30 nt were defined as clean reads and mapped to the mouse genome using meRanRh from meRanTK (version 1.2.1) [34]. Analysis of the unmethylated Xef mRNA spike-in controls were used to estimate the global bisulfite conversion rate. Unambiguously aligned reads were used to call candidate m^5^Cs by meRanCall from meRanTK. Sequential filtering steps were applied to ensure high-confidence in methylation calling: (1) 3C filter: bisulfite converted reads with more than 3 unconverted Cs were excluded. (2) Standard filter: coverage depth ≥ 20, methylation level ≥ 0.1, and methylated cytosine depth ≥ 6. (3) Signal/noise (S/N) filter: the m^5^C sites with “signal/noise” ratios greater than 0.9 were retained [8,35]. (4) False discovery rate (FDR) filter: adjusted *p*-values less than 0.05 were retained [12,34]. (5) Secondary structure filter: RNAfold of the ViennaRNA v2.2.9 software (–maxBPspan 150, -T 70, –MEA 0.1) was applied to predict conversion-resistant regions [36]. m^5^C sites located in these regions were further removed. After all the filtering steps, the remaining m^5^C sites that were present in both biological replicates were considered high-confidence m^5^C sites.

### 2.10. Determination of m^5^C Distribution, Differential Methylation and Correlation Analysis

The m^5^C sites were annotated using the GTF file from Ensemble. The m^5^C sites located within mRNAs were assigned into three segments: 5′ UTR, CDS, and 3′ UTR. Based on the ratio of the average lengths of 5′ UTR, CDS, and 3′ UTR in the transcriptome, we assigned 5, 22, and 18 bins to 5′ UTR, CDS, and 3′ UTR, respectively. The number of m^5^C sites located in each bin was counted and the percentage of m^5^C sites in each bin was calculated to plot the density of m^5^C sites along the mRNA transcripts.

The sites used for differential methylation analysis required the following two criteria: (1) coverage depth ≥ 10 in all replicates, and (2) high-confidence m^5^C sites in at least one condition. A customized Perl code implemented with the Fisher Exact Test was used to evaluate the significance of differential methylation, and the false discovery rate (FDR) method was used to correct for multiple comparisons. Sites with an adjusted *p* value ≤ 0.05 were considered as differentially methylated sites (DMS). The odds ratio (OR) or methylation fold change was calculated as previously described [37]. Pearson correlation between log2 expression fold changes and methylation change was performed to identify the correlation between RNA methylation and RNA expression.

## 3. Results

### 3.1. Robust Transcriptional Dynamics Induced by Neuronal Depolarization

To investigate mRNA methylation and expression upon neuronal activation, we performed both RNA-seq and RNA BS-seq with a widely used neuronal depolarization model (Figure 1a). Briefly, E16.5 mouse cortical neurons were dissociated and cultured in vitro as previously described [27]. To characterize the in vitro neuronal culture, we performed immunostaining using a neuronal marker (Tuj1) and a glial marker (GFAP). Immunostaining of the neuronal culture showed a dominant Tuj1 signal and a minimal GFAP signal, indicating high purity for the neuronal population (Figure 1b). After culturing in vitro for 7 days, the cortical neurons were depolarized with 55 mM KCl for 0 h, 2 h, and 6 h. At each end point, neuronal cells were harvested for total RNA extraction. After two rounds of poly(A) selection, the poly(A)-enriched RNA samples were subjected to RNA-seq and RNA BS-seq library construction. With two biological replicates for each condition, six RNA-seq and six RNA BS-seq libraries were generated and sequenced on the Hiseq 4000 platform with the 150 bp paired end mode.

We obtained comparable sequence coverages for the six RNA-seq libraries (Appendix A). For each library, an average of 32 million raw read pairs were generated with around 27 million read pairs uniquely mapped to the mouse reference transcriptome (Appendix A). For each time point, the two biological replicates showed similar expression profiles and were grouped together in the principal component analysis (Appendix A). Pearson correlation analysis also confirmed the high reproducibility between biological replicates (Appendix A). The R package DESeq2 was used for differential gene expression analysis [31]. With stringent cutoffs of fold change > 2 and adjusted *p*-value ≤ 0.05, hundreds to thousands of differentially expressed genes (DEGs) were determined upon neuronal activation. Specifically, we identified 583 up-regulated genes and 972 down-regulated genes in the comparison of 0 h vs. 2 h (Figure 2a,d), 1517 up-regulated genes and 1712 down-regulated genes in the comparison of 0 h vs. 6 h (Figure 2b,e), and 1573 up-regulated genes and 1294 down-regulated genes in the comparison of 2 h vs. 6 h (Figure 2c,f).

Gene Ontology (GO) annotation was performed to annotate the biological functions of these differentially expressed genes (Appendix A). Compared to the control (0 h), genes up-regulated at 2 h were highly enriched in transcription factors, such as *Egr1*, *Npas4*, and *Fos*, while genes up-regulated at 6 h were highly enriched in protein phosphorylation and neuron projection extension (Figure 2g). Interestingly, genes down-regulated at 2 h were enriched in a distinct group of transcription factors, including 161 Zinc-finger protein transcription factors (ZFP TFs). In the comparison between 2 h and 6 h, genes down-regulated at 6 h were enriched in DNA repair, while genes up-regulated at 6 h were enriched in a group of transcription factors that consisted of a large number of ZFP TFs. This indicates that the down-regulated ZFP TFs at 2 h were recovered later at 6 h. KEGG pathway analysis showed that genes up-regulated in the comparisons of 0 h vs. 2 h, and 0 h vs. 6 h were enriched in the MAPK signaling pathway, while genes up-regulated in the comparisons of 0 h vs. 6 h, and 2 h vs. 6 h were enriched in axon guidance (Appendix A). The above results collectively reveal the highly dynamic regulation of gene expression during neuronal depolarization.

### 3.2. Distinct Gene Expression Patterns in the Early and Late Phases of Neuronal Activation

To further decipher the distinct gene expression profiles observed upon neuronal activation, we pooled the six DEG lists identified in pair-wise comparisons and performed cluster analysis. Based on Z-score normalization, the DEG pool of 4736 genes was grouped into four clusters with distinct expression patterns upon neuronal activation (Figure 3a): decreased gene expression (cluster 0: 1794 genes), increased gene expression (cluster 3: 1086 genes), decreased gene expression at 2 h but increased gene expression at 6 h (cluster 1: 1329 genes), and increased gene expression at 2 h but decreased gene expression at 6 h (cluster 2: 527 genes). Sown-regulated genes (cluster 0) were enriched in DNA repair related pathways, such as non-homologous end-joining, DNA replication, base excision repair, and homologous recombination (Appendix A). This suggests that DNA replication and repair might be suppressed during the neuronal activation process. In contrast, genes that were up-regulated (cluster 3) were enriched in synaptic functions, such as chemical synaptic transmission, neuron projection extension, and axon guidance. Notably, genes in both cluster 1 and cluster 2 were enriched in DNA-templated regulation of transcription. Genes in cluster 1 were enriched in RNA modification, methylation, and tRNA processing. Genes in cluster 2 were enriched in the MAPK singling pathway and cell response to calcium ion. Such distinct functional enrichments among the four clusters indicate highly complex regulatory networks in activated neurons.

Cluster 2 and 3 may contain early and late response genes, which are well known to play key roles in processing external stimulation [27,28,38]. In this study, early response genes were defined as: up-regulated at 2 h compared to 0 h while down-regulated at 6 h compared to 2 h. On the other hand, late response genes were defined as: up-regulated at 6 h compared to 0 h and up-regulated at 6 h compared to 2 h. A total of 82 early response genes and 980 late response genes were identified (Figure 3c,d). The majority of these genes were included in cluster 2 or 3 (Figure 3e). The overlapped gene sets were considered as early response genes or late response genes with high-confidence and subjected to GO annotation analysis (Figure 3f). Early response genes were highly enriched in biological processes associated with transcriptional regulatory complexes, such as regulation of transcription from RNA polymerase II promoter and DNA-templated regulation of transcription, while the late response genes were associated with protein phosphorylation, transmembrane transport, and synaptic functions, such as regulation of membrane potential, chemical synaptic transmission, and axon guidance (Figure 3f). This result indicates that genes up-regulated in the early stage regulate the signal transduction cascade, while genes up-regulated in the late stage regulate the structural changes of neurons.

### 3.3. Transcriptome-wide mRNA m^5^C Modification in Mouse Cortical Neurons

Considering that genes down-regulated at 2 h were enriched in RNA modifications, we hypothesized that mRNA m^5^C methylation was dynamically regulated upon neuronal activation. To test this hypothesis, we performed bisulfite sequencing to obtain RNA BS-seq datasets. For the six RNA BS-seq libraries, an average of 97 million raw read pairs were sequenced with around 63 million read pairs uniquely mapped to the reference genome (Appendix A). To assess the overall bisulfite conversion efficiency, we included an in vitro transcribed Xenopus elongation factor 1α (Xef) mRNA as an unmethylated control. The overall conversion rates (C to T conversion) were estimated to be above 99.9% for all six RNA BS-seq libraries. To reduce the false positive methylation sites, several filtering steps were performed in the following order: (1) 3C filter: reads with more than 3 unconverted Cs were excluded; (2) a standard filter was applied with a coverage depth  ≥  20, methylation level  ≥  0.1 and number of methylated cytosine  ≥  6; (3) sites with signal/noise ratio (S/N ratio) below 0.9 were filtered; (4) sites located within RNA secondary structures were further filtered; and (5) false discovery rate (FDR) filter: adjusted *p*-values less than 0.05 were required to ensure statistical robustness.

High reproducibility of RNA BS-seq data was observed between biological replicates. The percentage of overlapped m^5^C sites between two biological replicates ranged from 30.88% to 52.64% (Appendix A), and the Pearson’s correlation for the methylation level of the overlapped m^5^C sites between two biological replicates ranged from 0.88 to 0.92 (Appendix A). The average methylation level of the overlapped m^5^C sites between biological replicates was higher than that of the m^5^C sites determined in one replicate alone (Appendix A). In this study, the overlapped m^5^C sites between the two biological replicates were considered as m^5^C sites with high-confidence and used for downstream analysis (Appendix A). In neurons stimulated with KCl, a total of 185 to 359 m^5^C sites within 157 to 314 RNA molecules were identified for (Figure 4a,b). The majority (97.50% ~ 98.77%) of these identified m^5^C sites were located within mRNAs (Appendix A). The remaining m^5^C sites were mapped to noncoding RNAs, including lincRNA, processed transcripts, and pseudogenes (Appendix A).

The majority of m^5^C sites (72.4% at 0 h, 73.5% at 2 h, and 61.7% at 6 h) had a methylation level below 30%, and only 6.7% to 7.6% of m^5^C sites showed methylation levels above 50%. The median methylation level of the m^5^C sites was approximately 20% among the three groups (20.7% in 0 h, 19.5% in 2 h, and 20.6% in 6 h) (Figure 4c and Appendix A). The sequence frequency logo showed a dominant G-rich triplet (A/G-GGG) motif downstream of the m^5^C sites (Figure 4d). A density plot showed a peak of m^5^C sites at the 5’UTR and an increasing peak at the translation initiation sites (Figure 4e). A large portion of the m^5^C sites identified (132 sites, 71.35% in 0 h, 36.77% in 2 h, 49.07% in 6 h) were shared among the three time points (Figure 4f). In all three groups, the m^5^C containing mRNAs showed enrichment of important neuronal functions, such as regulation of synaptic plasticity, synaptic transmission, synaptic vesicle exocytosis, and axon guidance (Figure 4g). This suggests that m^5^C modification may play a critical function in neurons. Interestingly, methylated mRNAs in neurons at 0 h and 6 h were enriched in chromatin organization, methylated mRNAs in neurons at 0 h and 2 h were enriched in memory, and methylated mRNAs in neurons at 2 h and 6 h were enriched in phosphorylation. The time point specific functional enrichments indicate that m^5^C modification is dynamically regulated during neuronal activation.

### 3.4. Dynamic m^5^C Landscape upon Neuronal Activation

To further determine the changes in mRNA m^5^C modification, we performed pair-wise comparisons of methylation profiles for the three groups (Figure 5a–c). Since substantial gene expression changes occur during neuronal activation, we limited our analysis to the m^5^C sites with sufficient coverage (20×) for comparison (Figure 5d–f). A total of 349 (0 h vs. 2 h), 266 (0 h vs. 6 h), and 372 (2 h vs. 6 h) m^5^C sites met this criterion. Compared to those at 0 h, the methylation levels of m^5^C sites in neurons at 2 h and 6 h were, in general, significantly higher. However, the methylation levels of m^5^C sites in neurons at 6 h were significantly lower than those in neurons at 2 h.

To further investigate the site-specific methylation changes, differential methylation analysis was performed. A total of 200 (0 h vs. 2 h), 108 (0 h vs. 6 h), and 136 (2 h vs. 6 h) differentially methylated m^5^C sites (DMS) were identified (Figure 5g–i). Compared to 0 h, 198 hypermethylated and 2 hypomethylated DMS sites were determined at 2 h, and 96 hypermethylated and 12 hypomethylated DMS sites were determined at 6 h. Compared to 2 h, 21 hypermethylated and 115 hypomethylated DMS sites were determined at 6 h. DMS-containing mRNAs from the three pair-wise comparisons were enriched in synaptic functions, such as regulation of synaptic transmission, regulation of synaptic plasticity, and axonogenesis (Figure 5j). More specifically, the DMS-containing mRNAs in the comparison between 0 h and 2 h were enriched in neuron cellular homeostasis, while the DMS-containing mRNAs in the comparison between 0 h and 6 h were enriched in protein transport and regulation of autophagy, and the DMS-containing mRNAs in the comparison between 2 h and 6 h were enriched in microtube cytoskeleton organization, cell migration, target of rapamycin (TOR) signaling and protein phosphorylation. These results suggest that the fine-tune regulation of m^5^C modification may occur upon neuronal activation to regulate the synaptic homeostasis.

### 3.5. mRNA m^5^C Methylation Negatively Correlates with Expression in Neurons upon Neuronal Activation

To further characterize the m^5^C methylation dynamics upon neuronal activation, we pooled the three lists of DMS sites for cluster analysis. According to Z-score normalization, the DMS pool was grouped into five clusters with distinct methylation patterns (Figure 6a,b): increased methylation at 2 h and then decreased at 6 h (cluster 0 with 90 DMS sites), increased methylation at 2 h and stayed hypermethylated at 6 h (cluster 1 with 51 DMS sites), decreased methylation at 2 h but increased at 6 h (cluster 1 with 13 DMS sites), increased methylation from 0 h to 6 h (cluster 3 with 40 DMS sites), and decreased methylation from 0 h to 6 h (cluster 4 with 9 DMS sites). GO annotation of the five clustered DMS-containing mRNAs showed distinct functional enrichments (Appendix A). Cluster 0 showed enrichment in synaptic functions, such as regulation of long-term synaptic potentiation, modulation of synaptic transmission, and regulation of axonogenesis. Cluster 0 also showed enrichment in the positive regulation of GTPase activity. Cluster 1 was enriched in the regulation of pre-synapse assembly and cell differentiation, cluster 2 was enriched in the regulation of synaptic vesicle exocytosis, cluster 3 was enriched in neuron projection development, and cluster 4 was enriched in the establishment of protein localization. In summary, opposite methylation trends may occur in neurons upon activation, such as cluster 0 vs. cluster 2.

Although DMS-containing mRNAs are critical for synapse plasticity in general, the functional enrichment results indicate that different trends in methylation changes are linked to genes responsible for diverse functions. Such dynamic methylation changes resemble the observed gene expression changes. This prompted us to further investigate the correlation between m^5^C modification and gene expression. We integrated RNA-seq and RNA BS-seq datasets to perform pair-wise correlation analyses for the changes in m^5^C methylation and the changes in corresponding mRNA expression. Consistent negative Pearson correlation between expression changes and m^5^C methylation difference was observed for all the three comparisons (0 h vs. 2 h: R = −0.19, *p*-value = 0.0083; 0 h vs. 6 h: R = −0.40, *p*-value = 2.1×10^−5^, 2 h vs. 6 h: R = −0.32, *p*-value = 0.00015) (Figure 6c–e). These results indicate that an increase in mRNA expression may result in decreased m^5^C methylation in activated neurons. We further categorized the transcripts carrying DMS into three different groups: positive correlation (hyper DMS with up-regulated DEG, hypo DMS with down-regulated DEG), negative correlation (hyper DMS with down-regulated DEG, hypo DMS with up-regulated DEG), neutral correlation (DMS only, not DEG). Genes with either negative correlation or positive correlation were determined (Figure 6f). Generally, more genes showed negative correlation, which is consistent with the global trend. Specifically, Fam122a, Zbed3 and Zfp553 showed consistent negative correlation in the comparisons of 0 h vs. 2 h, and 2 h vs. 6 h, and Rhbdd2 showed consistent negative correlation in the comparisons of 0 h vs. 6 h, and 2 h vs. 6 h. On the other hand, Homer1 showed consistent positive correlation in the comparisons of 0 h vs. 2 h, and 2 h vs. 6 h, and Sik3 showed consistent positive correlation in the comparisons of 0 h vs. 6 h, and 2 h vs. 6 h. This indicates the correlation between m5C methylation and mRNA expression could be transcript-dependent.

## 4. Discussion

In this study, we applied a classical neuronal depolarization model to investigate the dynamics of mRNA m^5^C modification in activated neurons. With both RNA-seq and RNA BS-seq datasets, we were able to profile gene expression as well as m^5^C modification at the transcriptome-wide level. With the RNA-seq dataset, we identified 4736 genes that were differentially expressed in response to neuronal activation. These genes were further clustered into groups with distinct expression profiles. It is not a surprise that genes involved in DNA replication or repair are down-regulated while genes involved in neuron projection and synaptic function are up-regulated in a stimulated neuron. In our study, many more late response genes were found than early response genes. This result is consistent with the concept that the early response genes often serve as transcription factors that regulate the expression of late response genes, which are involved in signaling pathways and diverse aspects of neuronal functions [27].

With the RNA BS-seq dataset, we identified a union set of 439 m^5^C sites with high confidence in neurons during different stages of neuronal depolarization. These m^5^C sites were supported by at least two biological replicates and passed a series of filters to remove potential false-positive sites using stringent parameters for sequence quality control and methylation calling. Notably, out of the 439 m^5^C sites identified, 132 (30.1%) were shared by neurons at all three time points. Several genome-wide RNA methylation studies indicated that RNA methylation gets its unique features differently from DNA methylation [7,8,12]. The levels of DNA methylation at CpG sites are often near 100% in mammalian somatic tissues, while only 20–30% of mRNA copies are methylated. In this study, the median methylation level of the m^5^C sites identified in neurons was approximately 20%. The majority of mRNA copies are unmethylated in neurons probably due to the high turnover rate of mRNA molecules.

Despite a large number of genes being differentially expressed, only a small set of transcripts experienced dynamic m^5^C modification upon neuronal activation. Interestingly, the methylation levels of 181 m^5^C sites were increased in neurons at 2 h post stimulation. The methylation levels of 40 sites continued to increase at 6 h, 51 stayed hypermethylated, and 90 sites returned to baseline at 6 h. Only 9 DMS sites had decreased methylation from 0 h to 6 h and only 13 DMS sites had decreased methylation at 2 h but increased at 6 h. GO annotation showed that DMS-related genes are significantly enriched in synaptic functions. This suggests that the m^5^C modification may play important roles in the regulation of synaptic adaptation in neurons in response to environment stimuli.

Furthermore, we investigated the relationship between RNA methylation and RNA expression. Pair-wise Pearson correlation was performed between gene expression changes and m^5^C methylation differences. Modest consistently negative correlation between RNA expression changes and methylation changes were observed. This finding suggests the regulation of RNA methylation may be linked to RNA expression. Altogether, our study provided transcriptome-wide maps of gene expression and m^5^C modification in neurons in response to environmental stimuli. Such a resource may lay the foundation for future studies to elucidate the role of m^5^C modification in regulating neuronal functions.

## 5. Conclusions

In conclusion, neuronal depolarization leads to drastic changes in gene expression together with changes in mRNA cytosine methylation. Our results indicated that mRNA cytosine methylation in neurons may play important roles in regulation of synaptic adaptation, and the regulations of RNA methylation and expression processes may be linked.

## Figures and Tables

**Figure 1 biology-11-00988-f001:**
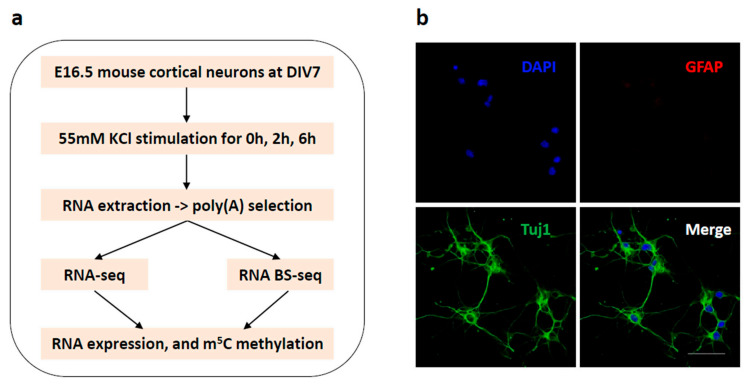
Experimental design. (**a**) Schematic representation of the experimental design. (**b**) E16.5 mouse cortical neurons were cultured in vitro for 7 days and double stained with the neuronal marker Tuj1 (green) and glial marker GFAP (red). Nuclei were counterstained with DAPI (blue). Scale bar: 50 μm.

**Figure 2 biology-11-00988-f002:**
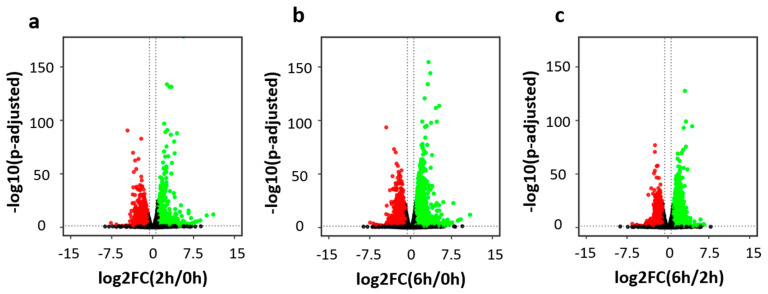
Neuronal depolarization induced dynamic gene expression. (**a**–**c**) Volcano plot showing the differentially expressed genes (DEGs) (fold change > 2 and adjusted *p*-value ≤ 0.05) in the comparison of 0 h vs. 2 h (**a**), 0 h vs. 6 h (**b**), and 2 h vs. 6 h (**c**). (**d**–**f**) Heatmap showing the gene expression level of down-regulated and up-regulated DEGs in the comparison of 0 h vs. 2 h (**d**), 0 h vs. 6 h (**e**), and 2 h vs. 6 h (**f**). (**g**) bubble plot showing significantly enriched GO terms of DEGs.

**Figure 3 biology-11-00988-f003:**
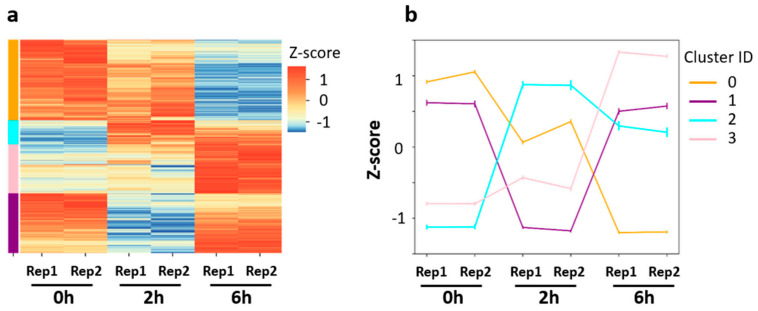
Characterization of the differentially expressed genes induced by neuronal activitydepolarization. (**a**) Clustering of the DEGs using Z-score normalization. The DEGs were grouped into four clusters. (**b**) Line plot showing the average expression level for each cluster. (**c**,**d**) Heatmap showing the gene expression profiles of manually defined early response genes (**c**) and late response genes (**d**). (**e**) Venn diagram showing the overlap between the clustering defined and manually defined early response genes (left) and late response genes (right). (**f**) GO annotation of the early response genes and late response genes.

**Figure 4 biology-11-00988-f004:**
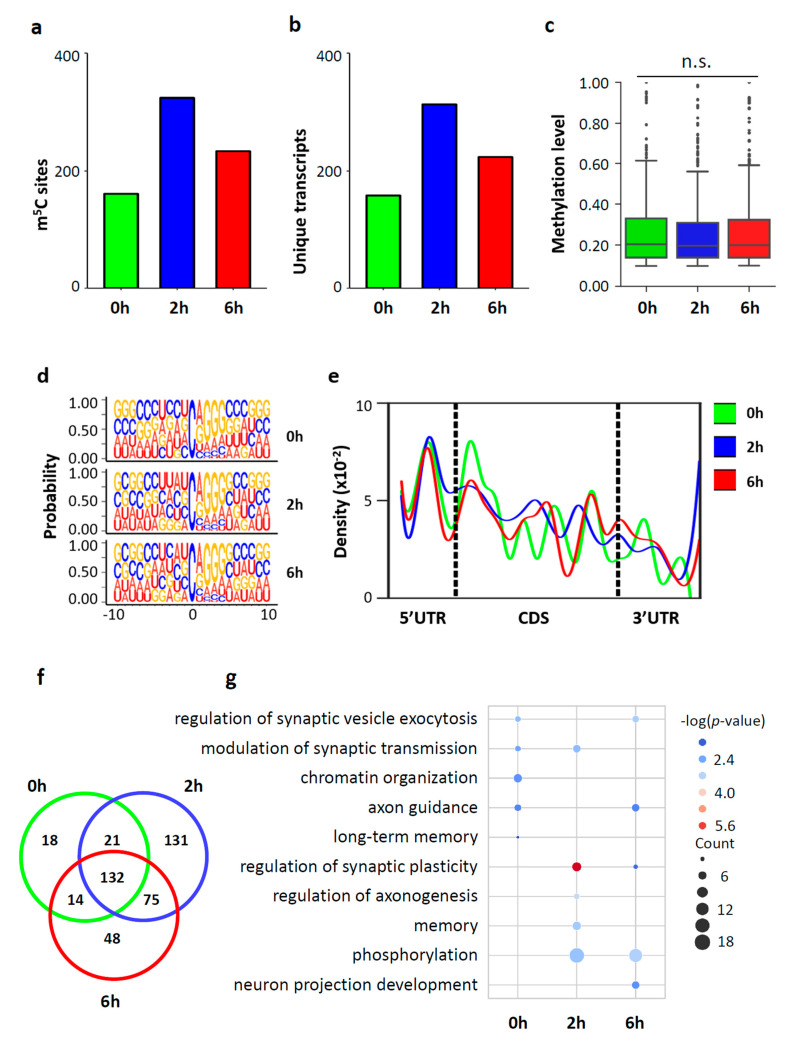
Distribution profile of m^5^C modification in mouse cortical neurons. (**a,b**) Bar charts showing the number of m^5^C sites (**a**) and m^5^C-modified mRNAs (**b**). (**c**) Boxplot showing the methylation levels of m^5^C sites. (**d**) Sequence frequency logo for the sequence context proximal to m^5^C sites. (**e**) Density plot showing the distribution of m^5^C sites along mRNA transcripts (5′UTR, CDS, 3′UTR). The moving average of percentages of mRNA m^5^C sites was shown. (**f**) Venn diagram showing the overlap of m^5^C sites among the three conditions (0 h, 2 h, 6 h). (**g**) Bubble plot showing the GO terms of m^5^C-modified mRNAs in neurons at 0 h, 2 h and 6 h.

**Figure 5 biology-11-00988-f005:**
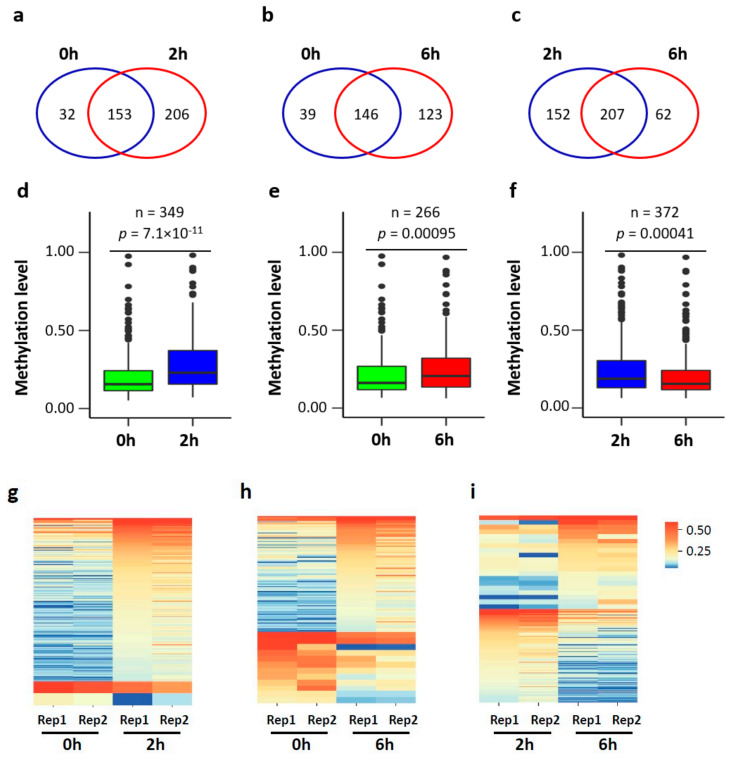
Neuronal depolarization induced m^5^C methylation changes in mouse cortical neurons. (**a**–**c**) Venn diagram showing the overlap of m^5^C sites between two conditions: 0 h vs. 2 h (**a**), 0 h vs. 6 h (**b**), and 2 h vs. 6 h (**c**). (**d**–**f**) Box plot showing the methylation level of the union of m^5^C sites between two conditions with 20x coverage in both samples: 0 h vs. 2 h (**d**), 0 h vs. 6 h (**e**), and 2 h vs. 6 h (**f**). A Wilcoxon signed-rank test was performed. (**g**–**i**) Heatmap showing the methylation profile of hyper- and hypo- DMS sites in the comparison of 0 h vs. 2 h, 0 h vs. 6 h and 2 h vs. 6 h. (**j**) GO annotation of transcripts with DMS sites.

**Figure 6 biology-11-00988-f006:**
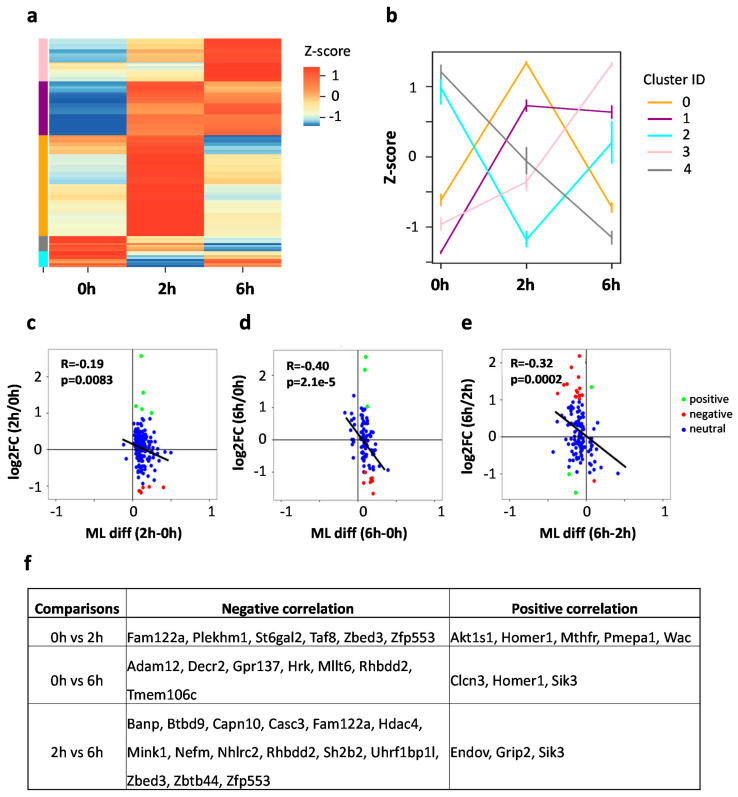
Correlation of RNA m^5^C methylation with RNA expression. (**a**) Clustering of the differentially methylated m^5^C sites (DMS) using Z-score normalization. The DMS sites were grouped into five clusters. (**b**) Line plot showing the average methylation level for each cluster. (**c**–**e**) Dot plots showing the pair-wise Pearson correlation between gene expression and mRNA methylation: 0 h vs. 2 h (**c**), 0 h vs. 6 h (**d**), and 2 h vs. 6 h (**e**). (**f**) A summary table showing the transcripts with negative/positive correlations between m^5^C methylation and mRNA expression among the three comparisons.

## Data Availability

Data generated in this study have been submitted to the NCBI Gene Expression Omnibus under accession number GSE201637. Analyses in this study were performed using the R v4.1.1, and Python 3.9.4 packages Biopython v1.78, matplotlib v3.3.4, Seaborn v0.11, and Pysam v0.16.

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
