# Peer review of "Neuronal Depolarization Induced RNA m5C Methylation Changes in Mouse Cortical Neurons"

_biology, 2022, doi:10.3390/biology11070988_

Round 1

Reviewer 1 Report

The authors are investigating RNA expression and mRNA methylation upon induced neuronal activity using KCl treatment of primary mouse cortical neurons.

My main concern, is the experimental setup. You are stating throughout the manuscript (including the title) that you are investigating changes in RNA expression and methylation upon induced neuronal activity. However, in no way are you actually showing that your KCl treatment results in neuronal activity.

KCl is widely known to cause de-polarization of neurons and also to cause a massive change in gene expression. However, at the dose used here, it has been reported that the de-polarization caused by KCl treatment do not result in increased neuronal activity.

To be able to state that the changes you see in gene expression and mRNA methylation is due to altered neuronal activity, you have to show that your KCl treatment actually induces neuronal activity, using patch clamp, MEA or other methods.

This impacts your statements both in the results and discussion section.

If this is not possible, you can only state that these changes are seen upon neuronal depolarization (not activity) and the manuscript has to be adjusted accordingly. However, this might reduce the novelty of the paper.

Author Response

We thank reviewers’ time, effort, and instructive advice.

Special thanks for pointing out the concern in the cell model. We agree with your opinion and have replaced "neuronal activity" with "neuronal depolarization" when necessary.

We implemented this in vitro neuronal activity model based on previous publications (1-3). The dose of KCl (55mM) used in this study is the same as the dose of KCl used in the previous two studies (2, 3). Stimulation of primary neurons with an elevated level of potassium chloride (KCl) leads to membrane depolarization and an influx of calcium through L-type voltage-sensitive calcium channels. The resulting increase in intracellular calcium level then triggers several calcium-dependent signaling pathways that ultimately lead to changes in gene expression (1).

This is considered as activity-induced transcription program and the signature genes include many immediate early genes (IEGs) such as c-Fos, Egr1, Nr4a1. Although we didn't perform patch clamp or MEA to show the direct evidence that the KCl treatment actually induces neuronal activity, all the signature genes (c-Fos, Egr1, Nr4a1) were significantly up-regulated at 2h in this study, which could be served as the molecular evidence of neuronal activation.

Gene expression level (TPM):

Gene

0h rep1

0h rep2

2h rep1

2h rep2

6h rep1

6h rep2

Fos

2.5

2.3

394.6

490.2

225.8

222.1

Egr1

4.4

4.7

22.6

24.1

10.1

9.3

Nr4a1

3.3

3.5

223.3

283.3

97.2

98.1

Reference

  1. Greer PL, Greenberg ME. From synapse to nucleus: calcium-dependent gene transcription in the control of synapse development and function. Neuron. 2008;59(6):846-60.
  2. Kim TK, Hemberg M, Gray JM, Costa AM, Bear DM, Wu J, et al. Widespread transcription at neuronal activity-regulated enhancers. Nature. 2010;465(7295):182-7.
  3. Joo JY, Schaukowitch K, Farbiak L, Kilaru G, Kim TK. Stimulus-specific combinatorial functionality of neuronal c-fos enhancers. Nature neuroscience. 2016;19(1):75-83.

Reviewer 2 Report

This is a beautiful written, conceptualised and presented manuscript, addressing m5C RNA methylation in the context of neuronal activity, correlating m5C signatures to transcriptional changes. Epitranscriptomic mechanisms turn out to be critical for brain development and function regulation, as well as in disease. I have nothing strong to criticise, just on aspect.The manuscript stops at an important point, which should be extended. For all datasets (RNA Seq and methylation analyses) the authors get very concrete in matters of pathways and genes being regulated, which I liked a lot. However, not so for the genes which are negatively correlated between expression and m5C methylation. Which pathways, GO terms or individual genes are affected? As obviously only 20% of the mRNA were methylated, a deeper look into the gene species would be informative. This should be added.  

Author Response

We thank the reviewer for raising this important point. We revised figure 6c,d,e by highlighting the genes with red color for negative correlation and green color for positive correlation. Also, we provided a summary table of gene list for negative and positive correlation in each comparison. The number of genes is pretty few, and no GO enrichment was detected. We added text in the results section discussing the individual genes that showed consistent correlation trend in more than one comparison.

Round 2

Reviewer 1 Report

The authors have addressed my concerns in a satisfying way, and I have no further comments on the manuscript.